# OpenReview forum: "Uncertainty-Aware LLMs Fail to Flag Misleading Contexts"
_NeurIPS.cc/2025/Workshop/Reliable_ML — NeurIPS 2025 - Reliable ML Workshop_

### Official Review · Reviewer_LtRc · 2025-09-08
**The paper claims that in context information influences the correctness and uncertainty of a model responses. The investigation was proved by a metric called EU. The result is that incorrect context can produce confidently incorrect responses.**

**Rating:** 6
**Confidence:** 3

**Review:**

## Summary
The paper claims that in context information influences the correctness and uncertainty of LLM responses. The investigation was proved by a metric called EU. The result is that incorrect context can produce confidently incorrect responses.

## Strengths
- This paper is particularly relevant to reliability with imperfect data, namely when the context provided to an LLM is incorrect, which is a common cause of confabulation.

## Weaknesses / Limitations
- In line 98 you use ChatGPT-4.1-mini to automatically rewrite the original supporting passages to introduce plausible but incorrect content. How did you verify that the newly generated content is plausible but incorrect after rewriting? For such a small model, it is likely to generate content that does not strictly meet your requirements, which is what you want to avoid in the first place.

## Suggestions for Authors
- Further expand on related works and how the current work connects with existing literature. Line 23 and 50 both mention evidential learning. Line 57 mentions Dirichelet based approaches. Perhaps these are worth explaining.
- Line 60 defines Au but is only used in Figure 4 in the appendix. Maybe you can briefly mention why this metric was not presented Section . Is there anything interesting that you can find using other metrics?
- In line 102, apart from prompting maybe you can also use vector similarity to check the semantic equivalence.
- Figure 2: You highlight two transitions but you can point out how the transition happens using the subplots as examples.
- In line 117 say what KDE means.
- In line 129, our earlier finding actually refers to line 113, which is not quite earlier finding. Maybe you can just move the paragraph.

---

### Official Review · Reviewer_jcYF · 2025-09-24
**Good exploration on failure mode of LLM**

**Rating:** 5
**Confidence:** 3

**Review:**

This paper studies how in-context information (accurate vs. misleading passages) alters LLM behavior and uncertainty estimates. The paper identifies an important failure mode of LLMs where correct context improves accuracy and reduces uncertainty, but misleading context induces confidently wrong answers.  While scope is limited, the work raises valuable questions for the community.

## Strengths
Problem motivation was clearly stated and it is tackling important failure mode of models. Broader impact of the observed problem (e.g., misinformation propagation) was well stated.

## Weaknesses and Suggestions
Although the paper tackles important problem and experiment was well structured, the evaluation is limited. As mentioned in the future directions section, including generalizability to open-ended generation or dialogue will strengthen the claim.

Would be great to see quantitative analysis on the relation between uncertainty and accuracy in relation to the degree of out-of-context input. For example, one can use counterfactual inputs of WOC:C items where they inject minimally altered counterfactuals. Analysis could be done on the knowledge-override rate vs. uncertainty shifts. This will be a good causal demonstration of context-induced overconfidence